# Enhancing Energy Efficiency of Sensors and Communication Devices in Opportunistic Networks Through Human Mobility Interaction Prediction

**DOI:** 10.3390/s25051414

**Published:** 2025-02-26

**Authors:** Ambreen Memon, Sardar M. N. Islam, Muhammad Nadeem Ali, Byung-Seo Kim

**Affiliations:** 1Information Technology Department, Torrens University, Melbourne, VIC 3000, Australia; drambreen.memon@torrens.edu.au; 2Institute for Sustainable Industries & Liveable Cities (ISILC), Victoria University, Melbourne, VIC 3000, Australia; sardar.islam@vu.edu.au; 3Department of Software & Communications Engineering, Hongik University, Sejong City 30016, Republic of Korea; nadeem@mail.hongik.ac.kr

**Keywords:** sensor communications, mobility encounter, similarity analysis, mobility, device-to-device communications, random forest regressor, opportunistic connectivity

## Abstract

The proliferation of smart devices such as sensors and communication devices has necessitated the development of networks that can adopt device-to-device communication for delay-tolerant data transfer and energy efficiency. Therefore, there is a need to develop opportunistic networks to enhance energy efficiency through improved data routing. A sensor device equipped with computing, communication, and mobility capabilities can opportunistically transfer data to another device, either as a direct recipient or as an intermediary forwarding data to a third device. Routing algorithms designed for such opportunistic networks aim to increase the probability of successful message transmission by leveraging area information derived from historical data to forecast potential encounters. However, accurately determining the precise locations of mobile devices remains highly challenging and necessitates a robust prediction mechanism to provide reliable insights into mobility encounters. In this study, we propose incorporating a random forest regressor (RFR) to predict the future location of mobile users, thereby enhancing message routing efficiency. The RFR utilizes mobility traces from diverse users and is equipped with sensors for computing and communication purposes. These predictions improve message routing performance and reduce energy and bandwidth resource utilization during routine data transmissions. To evaluate the proposed approach, we compared the predictive performance of the RFR against existing benchmark schemes, including the Gaussian process, using real-world mobility data traces. The mobility traces from the University of Southern California (USC) were employed to underpin the simulations. Our findings demonstrate that the RFR significantly outperformed both the Gaussian process and existing methods in predicting mobility encounters. Furthermore, the integration of mobility predictions into device-to-device (D2D) communication and traditional internet networks showed potential energy consumption reductions of up to one-third, highlighting the practical benefits of the proposed approach. The contribution of this research is that it highlights the limitations of existing mobility prediction models and develops new resource optimization and energy-efficient opportunistic networks that overcome these limitations.

## 1. Introduction

Smart devices such as mobile phones and laptops have gradually become everyday commodities in recent years due to their enhanced computing power. The processing and sensing capabilities of these smart devices have profoundly impacted various aspects of society [1]. They play a key role in expanding communication system coverage, becoming an integral part of the infrastructure through internet connectivity. The proliferation of billions of smart devices has generated a vast amount of data, presenting challenges in handling and transmission, commonly referred to as big data [2]. The transmission of such data has been a significant research area, with many studies focusing on developing various architectures for reliable, efficient, and resourceful designs. Among these architectures, the opportunistic network (Opp-Nets) stands out. Opp-Nets exploits time-delay-bounded file transmission using device-to-device (D2D) communication, rather than relying solely on the traditional internet infrastructure. In Opp-Nets, communities are established to route data from one node to another by defining the role of each node within the community [3]. These roles are assigned to various parameters, such as node popularity and centrality. The study of smart and connected communities lies at the intersection of civil infrastructures and cyber systems, offering a rich field for further research. There is existing research on and technologies involving algorithms such as the Gaussian process. These algorithms provide useful information and insights into the pattern of user movements. However, they are not fully accurate in real-life uses and complex cases. Complex cases reflects the trends and patterns in human mobility. In contrast to urban traffic, which has a hierarchical structure, human mobility possesses a high-level mobility trend comprising spatial, temporal, individual, and regional features [4] such as high-dynamic-mobility cases. The objective of this study is to develop an improved predictive model that can forecast human mobility interactions accurately and provide energy efficiency and improved network performance.

The rapid transformation driven by innovations in intelligent sensors is now embedded in nearly every physical device and system. The results of such advancements can be seen in several areas, including travel, energy [5], emergency management [6,7], and healthcare [8]. A network of mobility devices exhibits uncertain behavioral patterns, which can be a critical factor in optimizing the data routing process [9]. Users equipped with smart and mobile devices demonstrate almost identical activities in their daily routines. These identical activities can be beneficial in forecasting the possible interaction of these smart devices, eventually representing the encounter of users utilizing these devices [10], as illustrated in Figure 1.

A university campus environment is depicted in Figure 1, where users frequently visit various locations such as campuses, residences, hospitals, and related facilities. Typically, the mobility of these users can be described at the micro-level. For instance, a student or professor might visit the campuses, residences, and sports facilities as part of their daily routine, spending a specific amount of time at each location. A student or professor generally visits the campus from 9 AM to 5 PM, then moves to a sports area from 5 PM to 8 PM, and finally returns to their residence area for the rest of the evening, repeating this routine the next day. Moreover, while on campus, they follow specific routes, such as attending classrooms, delivering or taking lectures, and meeting colleagues. During these activities, they may encounter various communication devices that facilitate connectivity and file sharing, particularly over an Opp-Nets, which is suitable for delay-tolerant files [11,12].

This routine-based mobility information is valuable for predicting user mobility and can be leveraged for file sharing over the network, potentially reducing energy consumption. On the left side of Figure 1, we present a flowchart of the proposed mechanism for predicting these encounters using the random forest encounter prediction model (RFEPM). The flowchart elucidates the overall operation of the RFEPM in encounter prediction. We also propose employing supervised learning techniques, including the random forest regressor (RFR) and Gaussian process modeling, to predict future encounters based on historical patterns of individual nodes. We measured the predicted encounters at various locations using the Gaussian and RFR methods. These tests contrasted the precision of predicted encounters in similar locations in different weeks of the month. The highest accuracy achieved by the RFR was for location 5 (library), with an accuracy exceeding 85%. A Gaussian process is defined by a mean function and a covariance function (or the kernel), while the random forest is a supervised learning algorithm that randomly creates and combines multiple decision trees into one forest.

In this research, we propose a generalized method of predicting future encounters using the RFR and compare it with the Gaussian process probabilistic classifier. We utilized data traces from the University of Southern California (USC) for our experiments. The traces are actual measurements taken from the USC campus, containing information about Wi-Fi associations and users’ profiles. The data encompass six different buildings on the campus and access points and includes information from both weekdays and weekends. It is crucial to design a policy for Opp-Nets that dictates how a node will move a message through the network to meet latency constraints as much as possible [13]. Predicting a device’s potential encounters is essential for implementing a successful opportunistic routing algorithm [14]. This paper proposes employing the random forest and Gaussian models for mobility prediction and utilizes the predicted outcomes in the energy consumption model to save energy in smart buildings. The simulation’s results demonstrate that the proposed RFEPM learns occupant mobility behavioral patterns related to energy consumption within the building. The main objectives of the RFEPM and Gaussian model (GM) are to reproduce mobility encounters accurately, forecast the energy usage of buildings, and detect possible carbon loss zones.

The main contributions of the paper are as follows:We incorporate the RFR to predict human mobility encounters to optimize the resource allocation and energy-saving mechanism based on human mobility prediction. Additionally, we also incorporate the Gaussian model to predict the human mobility encounter as a benchmark scheme to compare the performance of both the RFR and the Gaussian model.We also highlight the advantage of energy consumption using device-to-device (D2D) communication in contrast to the utilization of energy consumption.By extensive simulations, we provide a comprehensive comparison of the proposed scheme with the existing schemes, which clearly illustrates that the proposed scheme outperforms the existing schemes.

The remainder of this paper is as follows: In Section 2, we present the literature review. Section 3 outlines the methodology, including the RFR, proposed RFEPM, the Gaussian model, the scenario for the RFR encounter prediction, and the energy consumption model for the sustainable energy approach. Section 4 describes the evaluation of the RFEPM, providing results for encounters of different users through the Gaussian model and encounter prediction results. This section also includes a comparison of the accuracy between the Gaussian model and RFR. Section 5 provides a comparison of the results and a discussion. Finally, Section 6 concludes the discussion and presents future work.

## 2. Literature Review

In this section, we will provide a comprehensive review of the contribution of existing work in designing various schemes to transfer data using the conventional infrastructure and device-to-device (D2D) communication mechanism incorporating human mobility information.

In today’s digitally revolutionized world, the number of smartphone users is increasing exponentially, resulting in a significant rise in mobile traffic [14]. Mobile devices form a wireless mobile ad hoc network (MANET), which offers various advantages in terms of scalability and flexibility. For instance, users can join or leave the network without any prior notification [15]. These mobile device users travel autonomously in different directions, allowing them to connect dynamically. This user mobility renders the MANET environment unpredictable, particularly for real-time data transfer applications [16,17]. It is important to note that opportunistic data transfer can be a potential characteristic of MANETs. However, the reverse may not always hold true, as not all MANETs rely on opportunistic behavior. MANETs can leverage opportunistic techniques for data transfer in delay-tolerant networks (DTNs), given that they primarily operate based on D2D communication. In [18], the author introduced the use of encounter history and transitivity for the prophet probabilistic routing protocol, based on the assumption of repeated rather than random movement. This approach results in quicker and more effective message delivery [19,20].

The probabilistic metric predicts an individual’s known destination by formally calculating each node before sending any message [15]. The probabilistic metric can be defined as P[a,b] in the range of [0,1] of two nodes, such as node A and node B. These nodes typically exchange probabilistic information with each other in a vector form upon each encounter [16]. Data are transferred to a neighbor node based on the high probability of them reaching the distinction; otherwise, the data are stored and transfer is refrained. The data caching in the ProPhet algorithm is implemented by using the FIFO queuing mechanism, which may result in the consistent dropping of the packets [17]. Another area for improvement in the protocols mentioned above is their slow and inefficient response to the dynamic trends in node movement. The main advantage of the protocol is that it uses the node’s delivery predictability information to choose the best carrier for message delivery [18].

ProPHET+ utilizes parameters such as buffer capacity, location, power, and popularity to select the optimal next hop. Normalized weights are assigned to these parameters using values from the range [0,1] [21]. These normalized weights and parameters are embedded in a metric to determine the next hop. Another routing protocol, based on historical information, was proposed by Boldrini et al. [22] for Opp-Nets. This protocol uses the current node context to select the ultimate path, which primarily consists of the identity table (IT) and history table (HT) to store the local context upon each encounter; additionally, the current context (CC) is updated. The collected information is monitored using continuity probability (Pc), redundancy (R), and heterogeneity counters. While HBOp stores all possible information, it faces memory capacity issues due to the requirements of the aforementioned counters.

Spyropoulos et al. [23] proposed an extended epidemic routing protocol known as the spray-and-wait protocol. This protocol primarily addresses congestion control caused by data flooding in the network. The proposed protocol operates in two phases: spray and wait. During the spray phase, the spread of the message is controlled by limiting its dissemination to specific nodes. In the wait phase, the relay node holds a copy of the message until it can directly transmit it to the destination.

## 3. Methodology

In this section, we will provide a detailed description of the proposed methodology incorporating the RFR and Gaussian model to predict human mobility traces.

Prior to the brief explanation of the proposed methodology, it is highly important to understand the significant potential of predicting human mobility. The core benefits of such prediction are resource optimization, such as the efficient utilization of constrained communication resources, and energy efficiency. Furthermore, the proposed methodology leverages the time–space association of the users over different time spans, which enables the robust understanding of the mobility patterns. Finally, the training of the dataset, which comprises various locations, such as a dataset of a university campus, could reflect the contextual analysis of the environment, resulting in the enhanced scope of the trained model [24,25,26]. This study incorporates human mobility patterns while prioritizing resource optimization and energy efficiency, as shown in Figure 2. We utilized the RFR, a reliable technique that works well with big datasets for encounter prediction. By comparing our method with a benchmark Gaussian model, we are able to illustrate a potential efficacy in reducing energy usage in smart connected buildings upon incorporating these predicted mobility encounters. To reduce energy consumption, the current system first utilizes the mobility traces of a user’s mobile phone, which can be conceptualized as a node in a communication network [27]. These traces provide time–space associations between the sender and receiver nodes, which are then used to measure the similarity rankings. Based on these similar profiles, the system predicts the future location and encounter time intervals for different users. It is also important to clarify that a trained model can be deployed on the network’s devices to predict a mobility encounter, forwarding the data to the next node.

In the final phase, an energy consumption model based on the opportunistic model is employed to reduce overall network energy consumption. Figure 2 illustrates the flowchart that depicts the RFEPM process for encounter prediction. When a user visits a new location, the system checks whether this location is included in the mobility preference of any other user. If no other user’s mobility preference aligns with this location, it is added to the user’s mobility history. The location ID and the duration of the visit are recorded in a new sub-decision tree to update the mobility history. Conversely, if a user with a mobility history that includes this location is identified, the visit time is compared to the visiting time of the first user. An encounter between these users is predicted at the specified location and time if a match happens [11]. Let us understand the basics of the RFR to clearly understand the proposed model.

### 3.1. Random Forest Encounter Prediction Model (RFEPM)

The RFR operates on the principle of decision trees, enabling decision-making based on specific conditions. Figure 3 illustrates the sample decision trees used to store the mobility histories of three users and predict their future encounters.

For example, Alice mostly visits the cafeteria on certain days and at certain times, followed by the library at a particular time. Similarly, Bob and Jane have their mobility traces stored in the decision tree. Each location visited by the users, along with the corresponding time, is recorded in the decision tree. This separate storage of each spatial-temporal pair of different users facilitates the retrieval of historical data and simplifies encounter predictions [3].(1)$$r_{j}=\frac{\sum_{j=1}^{n}o_{j}1\left[u_{j}\in C_{n}\left(U,E\right)\right]}{\sum_{j=1}^{n}1[u_{j}\in C_{n}\left(U,E\right)}$$
The RFR predicts future encounters by averaging the predicted future location of each user. We begin by selecting parameters suitable for our dataset, which are intended to enhance the accuracy of the RFR. Consider a training sample data input feature, $F_{n}=\left(F_{1},F_{2},\cdots F_{n}\right)$, where each feature is a random variable taking values in the interval [0,1]. We assume that these random variables represent encounters, with (e≥2) being the number of encounters, and each pair is independently and identically distributed. For fixed u∈[0,1], our objective is to estimate the classification function r(u)=E[F|u] using the feature $F_{n}$. Formally, a random forest is a predictor that comprises an ensemble of randomized base decision trees, $r_{n}\left(u,E_{m},F_{n}\right),m_{1}$, where $E_{1},E_{2},\cdots$ denote the outputs of a random variable, *E*. These random trees are generated to form the overall classification estimate. Specifically, we define $r_{n}\left(u,F_{n}\right)=E_{E}\left[r_{n}\left(u,E,F_{n}\right)\right]$, where $E_{E}$ denotes the expectation with respect to the random parameter *E*, conditional on the user *u* and the features of the dataset $F_{n}$. For simplicity, in the subsequent discussion, we will omit the explicit dependency on the sample and use $r_{n}\left(U\right)$ to denote $r_{n}\left(u,F_{n}\right)$. The RFR for encounter prediction consists of random decision trees. Each jth random tree, $r_{j}$, utilizes a coordinate, *U*, and variable, *E*, to determine the decision rule for partitioning data, as shown in Equation (Equation 1).(2)$$u_{n}\left(U,E\right)=\sum_{j=1}^{n}O_{j}1\left[u_{j}\in C_{n}\left(U,E\right)\right]$$
Here, *n* represents the sample size, and $u_{n}\left(U,E\right)$ denotes the time or event that determines the cut points for constructing the new tree, as illustrated in Equation (Equation 2). Let $C_{n}\left(U,E\right)$ represent the cell obtained after each random partition. Where $O_{j}$ is the output of the tree, assume that the data are spread in an Euclidean space and each random tree explores a sub-section of that space. Each tree addresses a specific region of the search space discretely and randomly. The selection of location sub-spaces is also a random process. This random selection of sub-space, when combined with the effect of bagging, enhances overall performance. The performance of each random tree is contingent on the selection of sub-spaces and the subsequent cuts applied in successive steps.(3)$$r_{j}\left(X\right)=E_{j}\left[r_{j}\left(U,E\right)\right]$$(4)$$E_{j}=E_{f}\frac{\sum_{j=1}^{n}o_{j}1\left[u_{j}\in C_{n}\left(U,E\right)\right]}{\sum_{j=1}^{n}1[u_{j}\in C_{n}\left(U,E\right)]}1\mu_{n}\left(“U,E”\right)$$(5)$$W=\sum_{i=1}^{p}R_{i}\left(U+E\right)$$(6)$$MAE=\frac{\sum_{i=1}^{n}|y_{i}-x_{i}|}{n}$$

Considering the final expectation $E_{f}$, the estimate for each random tree can be measured using Equation (Equation 4). Finally, the correlation among different trees is calculated using Equation (Equation 5).

Finally, the correlation among different trees is calculated using Equation (Equation 5). The mean absolute error (MAE) is calculated using Equation (Equation 6), in which $x_{i}$ is the encounter prediction made for each location, $y_{i}$ is the target encounter value, and *n* is the total number of places for which encounters are predicted.

### 3.2. Gaussian Model

The existing work on encounter prediction primarily employs the Gaussian model. The Gaussian model utilizes random variables that consist of a finite set, each possessing a joint Gaussian distribution. This model can also be characterized as a distribution over functions. The classification performed by the Gaussian process is non-parametric [28,29]. This method is predominantly used for opportunistic mobile networks (OMNs), where connections are sparse. The Gaussian process is represented through a covariance or mean function (also known as the kernel), where m(X) is the mean function. The covariance function k(x,x‘) for a real process, f(x), is illustrated in Equation (Equation 7). The Gaussian model for encounter prediction is also presented in Figure 4.(7)$$F\left(x\right)∼GP(m\left(X\right),\left(X,X^{\prime }\right))$$

The Gaussian model requires a matrix, $X=[x_{1},x_{2},x_{3},\cdots ,x_{n}]$, for training, with the dimensions (n,d) to represent the data points. Additionally, the model requires a vector, *Y*, denoted as ${\left[y_{1},y_{2},y_{3},\cdots ,y_{n}\right]}^{T}$, for the training dataset, with the dimensions (n,l). *f* represents the latent function, where $f(x_{i})$ indicates the predictive class membership probability. The Gaussian model maps the latent space to the observation space, a process that is usually non-linear. The kernel function is employed to transform a non-linear input space into a space represented by various dimensions, allowing the problem’s output to be expressed linearly. The significant challenge of this model arises when predicting the probability for a given test point, *X*, where *X* is a member of one of the specified classes.(8)p(y=C+|x)=sig(f(x))

The positive-class membership probability p(y=C+|x) is represented using Equation (Equation 8). The latent function f:R→R is mapped in the interval [0,1] through a sigmoid function, denoted as sig:R→[0,1]. Therefore, sig is a sigmoid transformation function and each data point, *x*, belongs to the finite dataset. Supervised learning is employed for classification, whereby each data point is assigned a class label. Specifically, each data point, *x*, is associated with a class label, $Y_{i}\in C^{+},C^{-}$, where $C^{+}$ represents the positive class and $C^{-}$ represents the negative class [30,31].

### 3.3. Energy Consumption Model for Sustainable Energy Approach

We developed a network model to calculate a more sustainable energy approach, representing our problem. Let us assume that G=(N,L) is a directed multi-network with *N* representing several nodes and *L* representing the set of edges in the network. In this network, edges represent the flow of objects from *i* to node *j* where i,j∈N. Edges (i,j)∈L have a capacity, and the associated energy consumption $C_{i,j},S\in N$ depicts the sender node and transmits to vertex x∈N. Bandwidth, defined as $u_{i,j}$, is also based on network link capacity. We consider demand and the source $B_{i}$ as follows: $B_{i}<0$ indicates a relay node with the demand $B_{i}=0$, and $B_{i}>0$ indicates a demand destination. For the prototype, an energy consumption model for both wireless and wired cases is considered. In this model, during the radio transmission generation, the transmitter and power amplifier consume power, and the receiver consumes energy to receive and process these radio transmissions.(9)$$E_{T_{x}}\left(k,d\right)=E_{T_{x}-elec}\left(k\right)+E_{T_{x}-amp}\left(k,d\right)$$(10)$$E_{T_{X}}\left(k,d\right)=\left\{\begin{array}{c} k\cdot E_{elec}+K\cdot \varepsilon_{f_{x}}\cdot d^{2},d<d_{0}\\ k\cdot E_{elec}+K\cdot \varepsilon_{mp}\cdot d^{4},d>d_{0} \end{array}\right.$$(11)$$E_{T_{x}}\left(k,d\right)=E_{R_{x}-elec}\left(k\right)=k\cdot E_{R_{x}-elec}$$

Taking radio transmission as an example, power control can be employed to mitigate signal propagation loss by appropriately adjusting the power amplifier. For instance, if the transmission distance is less than the threshold $d_{0}$, the free space prorogation model with the attenuation parameter $\varepsilon_{f_{x}}$ is used. Otherwise, the multi-path propagation model with the attenuation parameter $\varepsilon_{mp}$ is utilized. The energy consumption model for transmitting k-bits of data to a receiver at a distance of *d* can be calculated using Equation (Equation 9). The transmitter’s consumed energy, denoted by $E_{elec}$, depends on several factors, including digital coding, modulation, and filtering signal processing procedures. Additionally, the energy consumed by the amplifier is influenced by the distance to the receiver and the acceptable bit-error rate. The energy consumption for the data received can be calculated using Equation (Equation 11). The volume of data *k* and the transmission distance *d* are critical factors in the energy consumption model. These factors significantly affect the overall consumption compared to the energy consumption of other electronic components. Transmission distance plays a crucial role in energy consumption; greater transmission distance results in higher energy consumption. It is essential to shorten the transmission distance to reduce energy consumption. In this context, we will explain the utilization of the minimum-cost flow problem for the different data volumes and assess energy consumption by transferring data between two locations and nodes.

## 4. Performance Evaluation

In this section, we will provide comprehensive details of the performance evaluation and simulation environment, followed by the dataset, training stage, and testing stage descriptions. We utilized the JupyterLab environment on the Windows 11 platform, with 11th Gen Intel(R) Core(TM) i7-11700 equipped with 16 GB of RAM.

### 4.1. Simulation Environment

#### 4.1.1. Dataset

To evaluate the proposed work, we used a dataset of traces from the University of Southern California (USC) [32]. These traces were real-time mobility measurements collected from the USC campus, which comprised timestamp information collected through VPN sessions. During recording the dataset, it was compulsory for each user to become connected to a VPN session prior to accessing wireless connection. This mechanism resulted in recording the maintenance of the user accessing the network. Campus environments were chosen for their comprehensive, high number of active users and abundance of location samples. The dataset traces were collected from six different buildings on the campus. The buildings included in the dataset consisted of a cafeteria (location 1), a computer lab (location 2), residential facilities (location 3), a sports area (location 4), a library (location 5), and a campus hospital (location 6). The dataset incorporated users’ profiles, including their IDs, along with information about Wi-Fi connections, such as IP addresses, device MAC addresses, and the date and time of each connection. Mobility traces were recorded over various days and months, covering both weekdays and weekends. The dataset contained user log information collected from the access points, capturing the activity of every user accessing the campus internet. The data were recorded at one-second intervals, resulting in a dataset comprising approximately 7,776,000 samples. However, training machine learning models on such a large dataset requires significant computational resources and time. To facilitate simulation, a random selection of 100,000 samples from each month was made, yielding a dataset of 300,000 samples. This dataset was then divided into two subsets: 70% (210,000 samples) was for training and 30% (90,000 samples) was for testing. After preprocessing the data and removing noise, we selected the traces for January only. The mobility traces in the dataset were then divided based on the location. The January traces were utilized and split into two parts for training and validation purposes. We deployed the Gaussian process and RFR for classification. Figure 5 illustrates several of the dataset’s samples with various features such as time, IP address, MAC address month, etc.

#### 4.1.2. Training Stage

The construction of each tree in the random forest was achieved by selecting a random subset of samples from the training data (D). Each tree in the RFR was trained independently by sampling from the data on a random basis. During the training process, samples and features were selected with replacements, which means that some samples or features may have been repeated during the training of each tree in the forest. In the training stage, the construction of each tree followed the procedure outlined below:Count samples in the randomly chosen sub-space $C_{n}\left(U,E\right)$.The goal of decision tree training is to reduce the entropy *S* and maximize information gain.For each selected feature, compute the feature’s information gain and evaluate the difference. For instance, for the encounter feature, compute the entropy and information gain.The coefficient of variation is employed to set a threshold for further branching. If a node contains fewer than a specific number of samples, the partitioning process stops, and the node is designated as a leaf node.

In this simulation, the mean square error (MSE) was used as a measure of classification error, where the term “samples” refers to the number of data samples represented by the node, and “value” denotes the average value of the data that the node represents. For this particular simulation, the minimum number of samples required in a node was set to one, which implies that a node could contain only one sample.

#### 4.1.3. Testing Stage

The model evaluated all the trees using the estimation function, which assessed each random tree individually. The majority vote from the outputs of all the trees was then used as the final predicted value. The accuracy of each tree was computed as the ratio of correct encounters to the total number of encounters in the data, considering only encounters that occurred at a specific point in time, where time is represented by (U,E). The aggregate prediction was obtained by synthesizing the prediction from all the trees.

#### 4.1.4. Validation of Output

To evaluate the location feature of the number of users in the randomly selected sampled data $C_{n}$ (data partition made through bootstrapping), we first recognize that location is considered the initial feature. However, it is essential to note that since features are selected in a random sequence, the day of the week or time might be chosen as the first feature in some decision trees. This randomness in feature selection contributes to the uncorrelation of different trees in the forest, which is a fundamental characteristic of random forests. In this context, $u_{j}$ represents one such randomly selected feature from the set. Additionally, (U,E) denotes the time information regarding the user’s presence at the chosen location, while $O_{j}$ refers to the output of the tree.

### 4.2. Simulation Results

In this section, we evaluate the proposed scheme for encounter prediction among users. First, we calculated the encounter results using both Gaussian and RFR algorithms. Subsequently, we computed the energy using the RFEPM results, as the RFEPM demonstrated superior predictive performance compared to the Gaussian model.

These results were calculated for five users considering the day of the week and the hour of the day. An encounter observed in a specific hour was denoted as 1, while the absence of an encounter was denoted as 0. To determine the encounters, we focused on the weeks in which an encounter occurred on the specified day and in the specified hour. Figure 6 and Figure 7 illustrate the encounters that occurred in January at the cafeteria and library, respectively. The *x*-axis represents the ordered pair (day, encounters occurred) for the month, while the *y*-axis shows the normalized values of the encounters. In Figure 6, the highest normalized encounters at the cafeteria in January occurred on days 6 and 24. The lowest number of encounters, which was zero, occurred on day 1. The peaks in the graph represent the varying encounter frequencies across different days. Similarly, at the library, as shown in Figure 7, the highest number of encounters, nineteen, occurred on day 8, while the lowest number of encounters, only one, occurred on day 11. Figure 8 and Figure 9 present the encounter predictions and the actual encounters observed through the Gaussian process for the cafeteria and library, respectively. In both Figure 8 and Figure 9, the actual encounters are illustrated using the red dotted line, while the predictions are illustrated using the blue solid line. In both Figure 8 and Figure 9, the timeline is shown on the *x*-axis, while prediction results are scaled in the range of [−0.2 1.0]. The cafeteria illustrates a higher number of encounters than the library, as most people tended to visit the cafeteria frequently.

Figure 10 and Figure 11 show the actual and predicted encounters through the RFR at the cafeteria and library, respectively. The encounters are illustrated using a red dotted line, while predictions are illustrated using a blue solid line in Figure 10 and Figure 11, respectively. The *x*-axis represents the total number of weeks across the entire dataset during which encounters occurred, while the *y*-axis displays the normalized values of encounters. Figure 10 and Figure 11 both illustrate the very close relationship between predicted outcomes and human mobility encounters. The RFR outperformed in prediction performance as compared to the Gaussian model in predicting human mobility encounters. The comparison of both incorporated schemes and a comparison with existing schemes is illustrated in Table 1.

To compare the prediction performance of the Gaussian and RFR methods, we assessed the accuracy of both models’ predictions across different locations through each approach. These analyses evaluated the accuracy of predicted encounters at a specific location over various weeks of the month. The graphical representation of accuracy comparison for results obtained through both approaches (Gaussian and random forest) is illustrated Figure 12. As shown in Figure 12, the RFR demonstrated a significantly higher accuracy compared to the Gaussian model. The RFR achieved its highest accuracy at location 5 (library), exceeding 85%. The RFEPM model offered valuable insight into human encounters. To validate our proposed methodology, we used a dataset of 200 users to compare the energy consumption between internet-based and D2D communication systems. As shown in Figure 13, although energy consumption increased with the number of users transmitting data in the network, it was still significantly lower compared to the energy required for data transfer via the internet. We thoroughly compared the Gaussian process and random forest algorithms using real-world mobility data traces and demonstrated that encounter predictions could effectively conserve resources. This study aims to reduce energy consumption during data transmission in everyday scenarios. Our RFEPM approach illustrates notable energy savings in smart connected buildings.

## 5. Results’ Discussion and Comparison

This section provides a comprehensive discussion of the proposed work’s performance and its comparison with existing schemes.

To evaluate the performance of the proposed approach, we consider several baseline methods, as summarized in Table 1. The existing schemes utilize statistical information from three cities, i.e., New York City (NYC), Tokyo (TKY), and Dallas, to analyze the human encounters across various users and locations. The discussion references mobility studies from New York, Tokyo, and Dallas to contextualize the results and benchmark against existing schemes. However, these datasets were not directly integrated into the USC analysis. Future iterations of this work will include a comparative evaluation of mobility patterns and model performance across cities, considering geographic and cultural differences to improve coherence and validity. For comparison, we selected a specific location where the highest prediction accuracy was observed.

The baseline algorithms used for the comparison are as follows:In [33], the Markov-based model known as FPMC primarily employs factorization to learn transition matrices, achieving the highest accuracy of 68.8% in Tokyo.DeepMove [34] uses an attention-based technique combined with a GRU to capture both long-term and short-term preference, achieving the best prediction accuracy of 58.6% in New York City.In [35], Flashback predicts human encounters in New York City with the highest accuracy of 73.7%, utilizing the spatio-temporal distance to derive hidden states of the current prediction.In [36], LSTPM employs temporal similarity and distance factors to learn the long-term preferences and geography relevance for short-term preferences, achieving the best accuracy of 76.5% in New York City.In [37], GeoSAN introduces a newly designed geography encoder to learn spatial proximity, attaining an accuracy of 60.2% in New York City.In [38], STAN implements a two-layer attention architecture to capture spatio-temporal correlations, achieving an accuracy of 59.1% in New York City.In [39], GETNextutilizes a graph convolution network (GCN) to learn collective movement patterns and employs a transformer encoder to capture transition regularities.In [40], CSLSL applies a causal and spatial-constrained long- and short-term learner for location prediction, achieving a prediction accuracy of 66.1% in the New York City dataset.

## 6. Conclusions and Future Work

This study develops an RFR that substantially enhances the accuracy of mobility prediction and energy efficiency in D2D communication systems. We utilized human mobility patterns to develop an efficient data transmission process that requires fewer resources, specifically in terms of energy and bandwidth. Our proposed model incorporates knowledge of energy usage and occupant behavioral trends within buildings. Additionally, we focus on encounter prediction, which involves forecasting when two or more users will converge at a specific location and time. Accurate encounter prediction can help to minimize resource consumption by reducing the need for continuous searches for potential encounters. For encounter prediction, we employ the RFR, which organizes historical human mobility traces into a tree structure. The model first predicts future location based on past mobility patterns and, using similar mobility preferences, forecasts encounters for a specific day of the month and hour of the day. We also developed a mathematical computational model to predict encounters between different users.

In our experiments, we trained the USC dataset using both a benchmark Gaussian model and the RFR. The results were analyzed for observed versus predicted encounters using both approaches. The accuracy of encounter prediction was compared, and the findings indicated that the RFR outperformed the benchmark Gaussian model. In fact, at some locations, the encounter prediction exceeded 85%. Our proposed method demonstrates that the RFR can effectively save energy in smart, connected buildings. Future research can expand the current model by integrating it with other emerging technologies, such as quantum computing and edge computing, and apply the expanded model to many uses in different mobility cases.

The reliance of Opp-Nets on user-installed and active applications or protocols is acknowledged as a limitation, as it may introduce biases in data collection and limit its generalizability. Future work will focus on integrating the proposed model with widely used existing networks, such as Wi-Fi or 5G, to ensure broader data collection and improved representativeness. Leveraging these infrastructures can mitigate the dependency on specific user-installed applications while enhancing the applicability of the findings in diverse real-world scenarios.

## Figures and Tables

**Figure 1 sensors-25-01414-f001:**
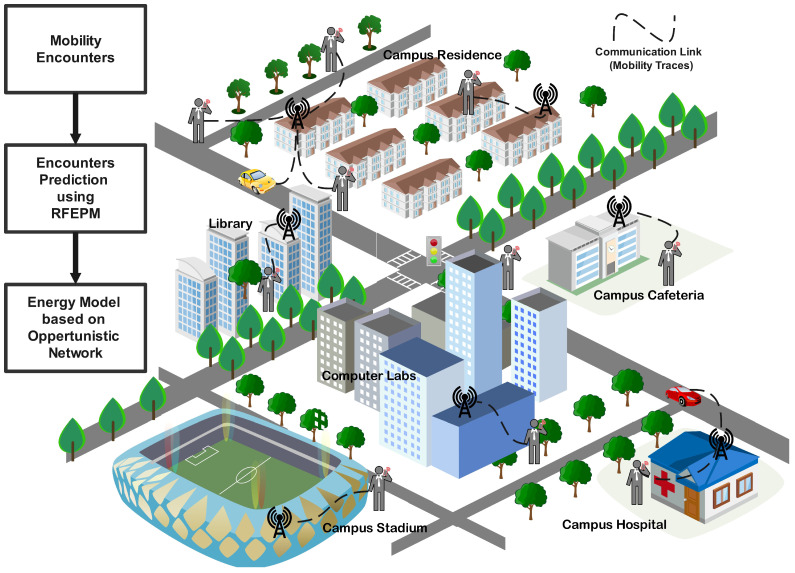
Internet infrastructure for the mobility scenario of the university campus.

**Figure 2 sensors-25-01414-f002:**
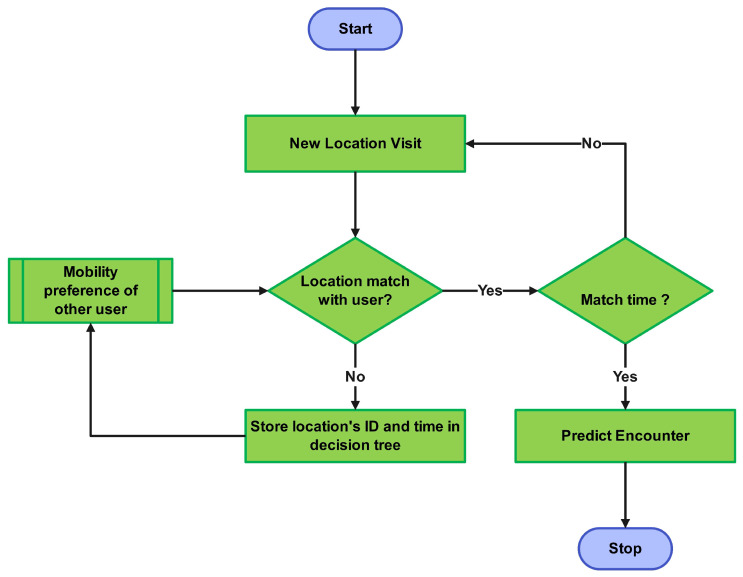
Proposed methodology flowchart.

**Figure 3 sensors-25-01414-f003:**
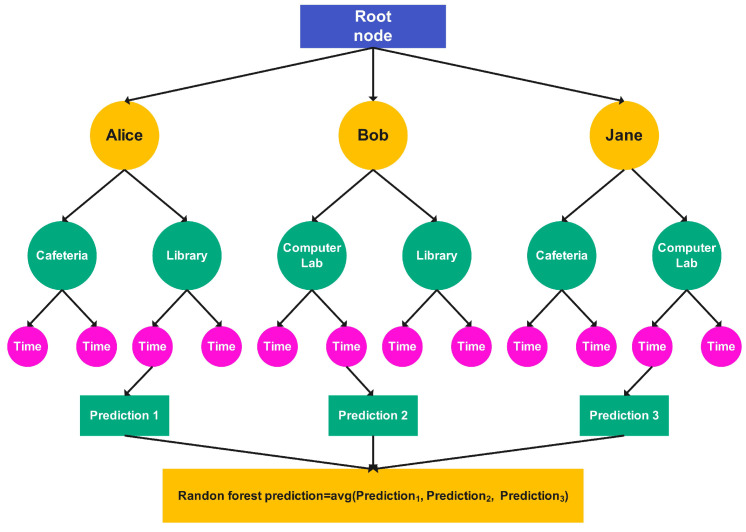
Encounter prediction by the RFR.

**Figure 4 sensors-25-01414-f004:**
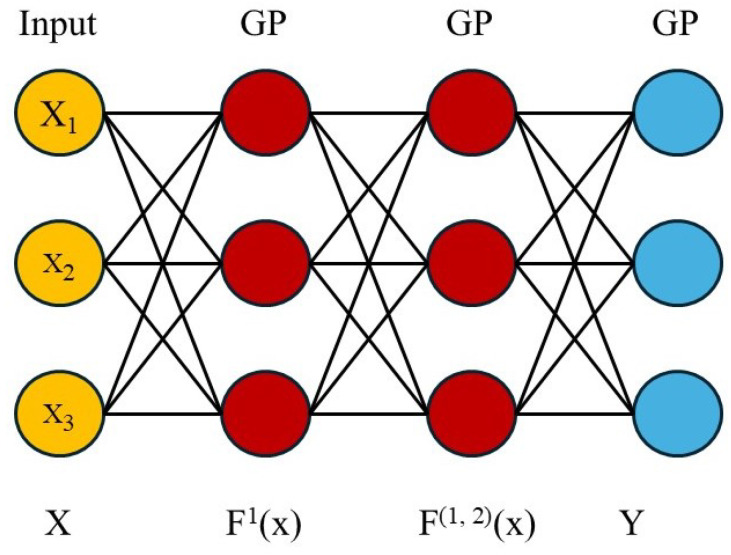
Gaussian model for encounter prediction.

**Figure 5 sensors-25-01414-f005:**
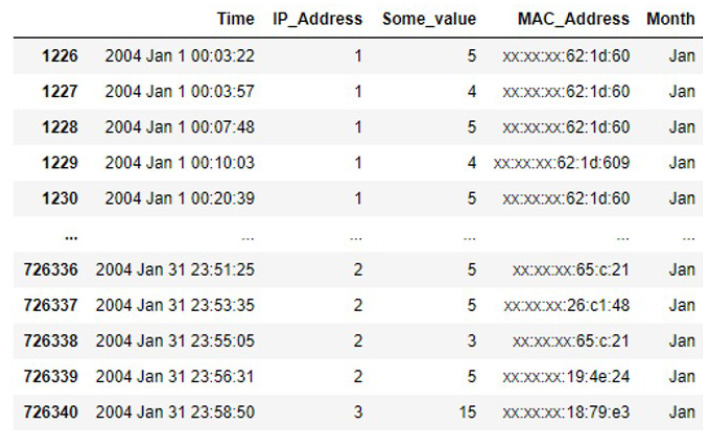
Some samples of the dataset.

**Figure 6 sensors-25-01414-f006:**
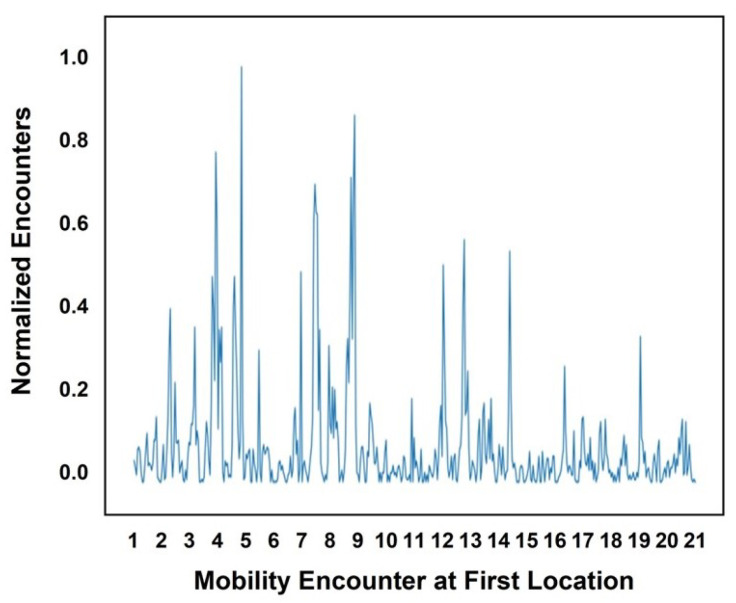
Encounters observed in the cafeteria.

**Figure 7 sensors-25-01414-f007:**
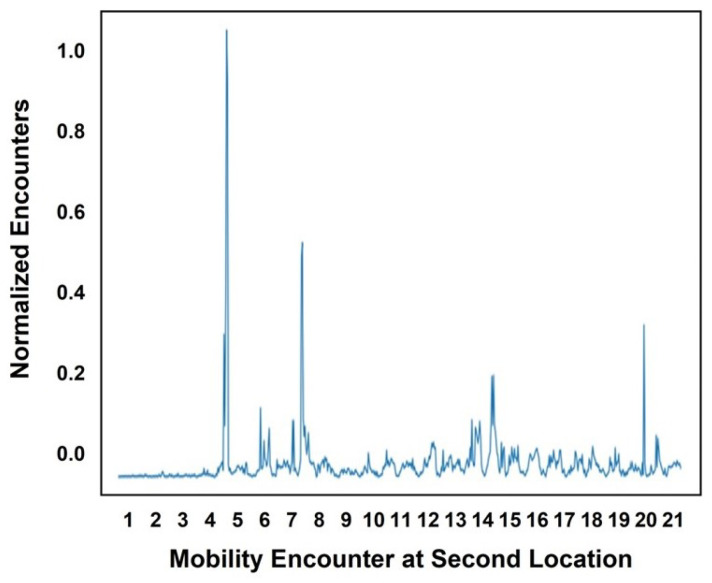
Encounters observed in the library.

**Figure 8 sensors-25-01414-f008:**
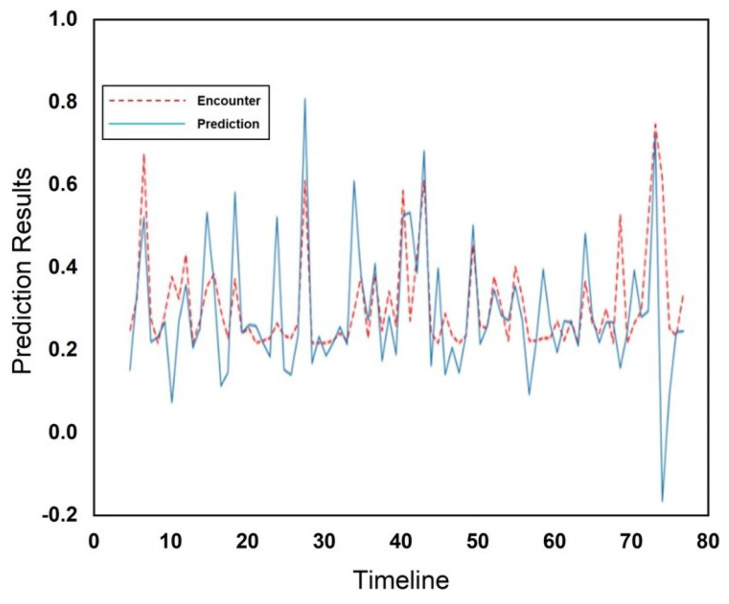
Encounter prediction using the Gaussian model for the cafeteria.

**Figure 9 sensors-25-01414-f009:**
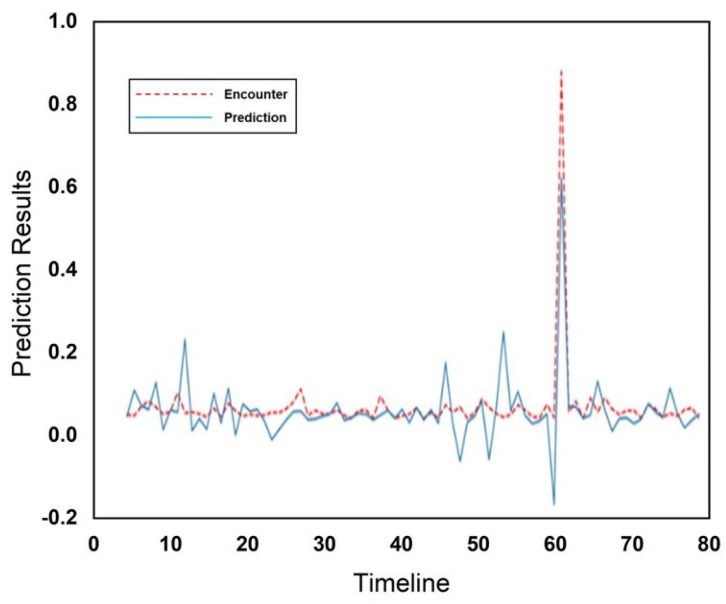
Encounter prediction using the Gaussian model for the library.

**Figure 10 sensors-25-01414-f010:**
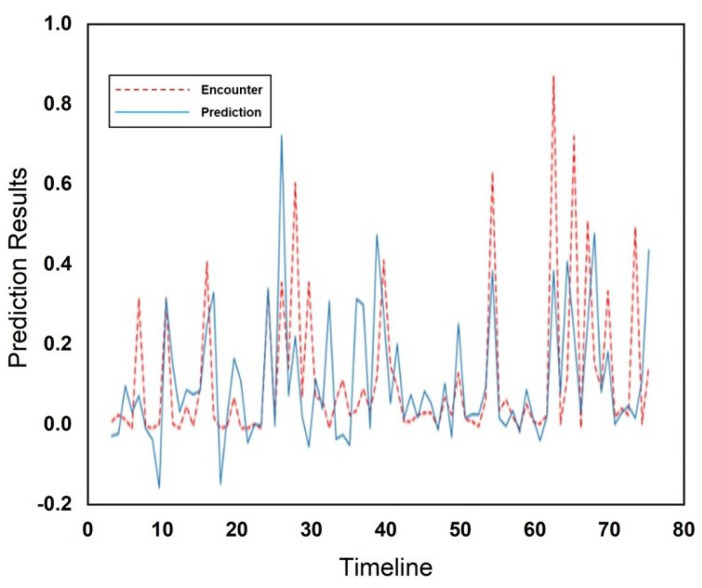
Encounter prediction using RFR for cafeteria.

**Figure 11 sensors-25-01414-f011:**
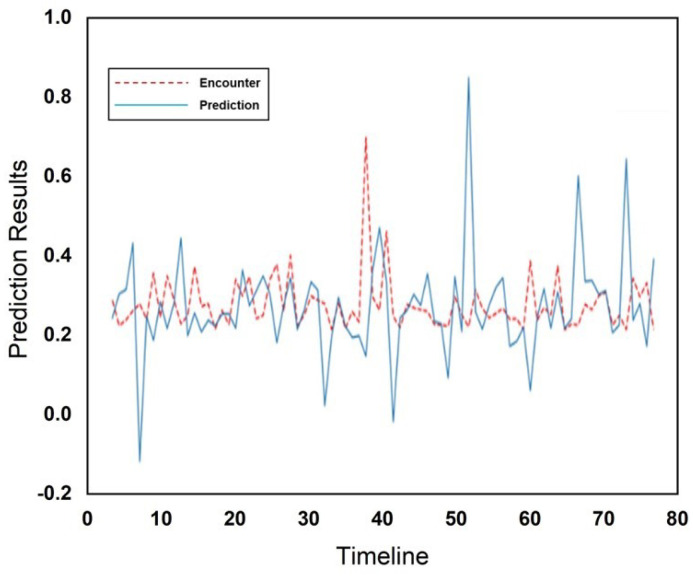
Encounter prediction using RFR for the library.

**Figure 12 sensors-25-01414-f012:**
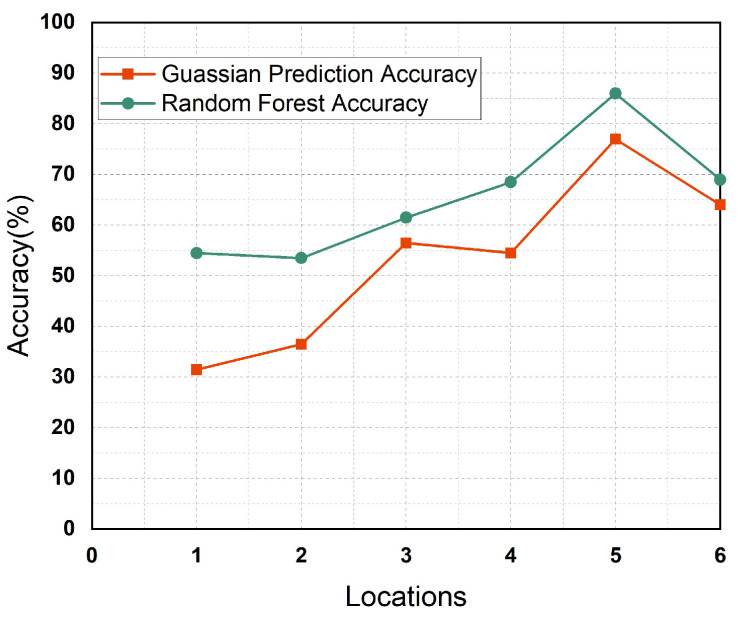
Random forest and Gaussian prediction.

**Figure 13 sensors-25-01414-f013:**
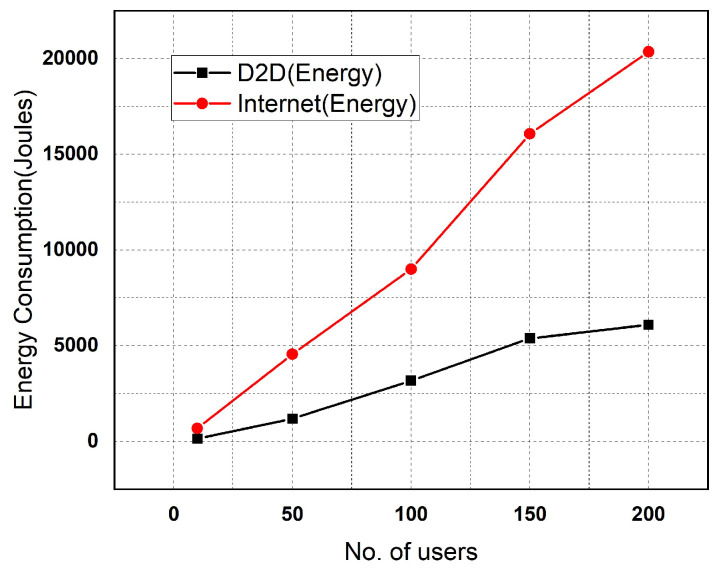
D2D vs. internet architecture energy consumption.

**Table 1 sensors-25-01414-t001:** Comparison with existing schemes.

Ref. No	Algorithm	Best Location	Accuracy
[33]	FPMC	TKY	68.8%
[34]	DeepMove	NYC	58.6%
[35]	Flashback	NYC	73.7%
[36]	LSTPM	NYC	76.5%
[37]	GeoSAN	NYC	60.2%
[38]	SATN	NYC	59.1%
[39]	GETNext	NYC	74.9%
[40]	CSLSL	NYC	66.1%
**Our**	**GP**	**location 5 (library)**	**77%**
**Our**	**RFR**	**location 5 (library)**	**86%**

## Data Availability

Wei-jen Hsu Hsu, Ahmed Helmy. (2022). CRAWDAD usc/mobilib. IEEE Dataport. https://dx.doi.org/10.15783/C79W25.

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
