# Peer review of "Enhancing Energy Efficiency of Sensors and Communication Devices in Opportunistic Networks Through Human Mobility Interaction Prediction"

_sensors, 2025, doi:10.3390/s25051414_

Round 1

Reviewer 1 Report

Comments and Suggestions for Authors

The authors present a new way of predicting motion for use in opportunistic networks. The structure of the paper is adequate, although some sections need to be improved. The explanation of the results should be more complete. The English sounds good. The references are up to date, but more should be added.

Some improvements are below.

1. Internet should be capitalized, since it is a proper noun.

2. Figure 1. It is not a city, it is a university campus in the USA.

3. Figure 1. The acronym RFEPM appears, but it has not been mentioned before the Figure. It is confused with the acronym RFM. This should be clarified before seeing the figure.

4. Table 1. It is not cited in the text. How does the system know the coordinates? Was a GPS used in the sampling?

5. Section 2. It must be related to what is proposed in the paper. How will the prediction be used in these algorithms? A MANET network is cited that is not opportunistic. More citations related to the topic of the article should be added, some from the year 2024 if possible.

6. Section 3. The acronym MAE is not defined. When the article is printed, some lines in figure 4 do not appear.

7. Figure 5. It is not cited in the text.

8. Section 4. More information about the dataset should be added: total number of samples, number of samples for training, number of samples for simulation... How have the validation and training sets been chosen? Who makes the prediction and when? Why are only 5 users used? Figure 13 is the most interesting, how has the D2D protocol been changed to add the prediction?

9. Section 5. I don't understand it. The authors, in order to compare their proposal with that of others, should apply it to the other datasets and see if it is improved or not. This section does not contribute anything to the article, it should be eliminated.

10. Section 6. How could the proposal be applied in real time? Could the prediction be changed with new data?

Reviewer 2 Report

Comments and Suggestions for Authors

Review sensors-3418073

Date: 01/05/2024

Title:
Enhancing Energy Efficiency of Sensors and Communication Devices in Opportunistic
Networks through Human Mobility Interaction Prediction

1. Original Submission
1.1 Recommendation
Reconsider after major revisions

2. Comments to Author:
Ms. Ref. No. sensors-3418073

2.1 Overall and general recommendation
The article proposes a Random Forest Model (RFM) for predicting human mobility encounters, focusing on optimizing energy efficiency in device-to-device (D2D) communication systems. The authors integrate human mobility patterns and spatiotemporal analysis to reduce energy consumption in connected smart buildings. The RFM is compared to a Gaussian reference model, with the former demonstrating superior predictive accuracy and energy savings.

The implementation of the RFM as a predictive model based on historical mobility data and its integration with D2D systems represents an interesting approach. It highlights the use of mobility patterns to minimize continuous searches and optimize resource allocation. The experiments show that the RFM outperforms the Gaussian model, achieving an encounter prediction accuracy of over 85% in some locations and demonstrating notable energy savings compared to other systems. While the dataset used (USC mobility traces) offers a realistic environment with highly active users and multiple locations, the study's focus is limited to the USC campus, making generalization of the results difficult. Additionally, the study does not explore different environments or mobility contexts, even as a limitation.
Overall, the article fails to evaluate how key parameters (e.g., dataset size, node
density, or variations in mobility patterns) affect the model’s performance.  Although several reference methods are presented, comparisons lack detailed metrics such as computational cost or training time for each approach.
The identified issues affect data representativeness, methodological clarity, study reproducibility, and result generalization. The lack of justification for key areas (e.g., choice of training month, inclusion of other cities) and omission of relevant information (e.g., building characteristics, antenna placement) lead to the recommendation for major revisions to improve the research.

2.2 Mayor comments
— One of the study’s core aspects is its reliance on mobility behavior patterns (Line 167). However, starting from Line 59, the proposed schedules are questionable in today’s
context, particularly in a post-pandemic era where mobility habits have become more dynamic. Factors such as hybrid classes, remote work, and shifts in user behaviors significantly influence mobility.

— The study is based on past mobility models (Line 443), yet it lacks a clear description of these patterns and how they were obtained. Ignoring such variability makes the model’s results potentially less effective in dynamic scenarios or environments where mobility patterns are not as consistent.

— The article could address how key variables (e.g., time, location) were identified to train the model. Additionally, it should consider whether mobility patterns observed in a university campus are representative enough to apply the methodology to other environments.

— Mobility patterns in a university campus vary due to factors such as exams, holidays, and weather conditions, none of which are analyzed in the article. For example, the cafeteria’s high encounter probability is expected, but no detailed analysis is provided. The proposed methodology could be a valuable tool for studying these variations, but this potential is not explored.

— The article does not specify the six buildings analyzed on the USC campus or their main characteristics (e.g., type, purpose, user density). This omission complicates understanding the study’s context and limits reproducibility. Figure 1 is overly simplified and does not align with the actual layout of the USC campus, creating issues for following the research and extrapolating results.

— Urban layout and building types (e.g., libraries, cafeterias) can significantly influence mobility patterns and encounter predictions. High-density buildings may generate more encounters, while specialized buildings (e.g., labs) may have more predictable patterns. These factors are not discussed.

— Opp-Nets require users to have an installed and active application or protocol, posing a significant limitation for mobility studies. This requirement could bias the collected data and limit its representativeness. The article does not address this limitation or propose mitigation strategies, such as integrating with existing networks (e.g., Wi-Fi or 5G), which could provide broader and more representative data.

— The choice of January as the sole training month is not justified. Mobility patterns can vary significantly throughout the year, and training the model with data from a single month limits its generalization. The article should include a comparative analysis with data from other months to assess the model’s stability and its ability to capture seasonal patterns.

— The inclusion of data from New York, Tokyo, and Dallas in the discussion (Line 406) is not contextualized or mentioned in the study design. It is unclear how these data were obtained, integrated with USC data, or whether differences between these cities were considered. This affects the coherence of the article and the validity of its comparisons.

— While the model shows promising results in terms of accuracy and energy savings, the article does not discuss its implementation in real-world systems, such as smart buildings or D2D networks. This lack of practical analysis limits the impact and relevance of the work.

— Although not the primary focus, the methodology could be a powerful tool for analyzing how mobility patterns change over time or in specific contexts. This potential is not mentioned in the article.

2.3 Minor comments

— Line 81: Reference to "Location 5" is missing from Figure 1.

— Line 47: Clarify what is meant by “Complex Cases.”

Round 2

Reviewer 1 Report

Comments and Suggestions for Authors

The authors have made most of the suggested changes, the paper could now be published.